# *De novo* Drug Design using Reinforcement Learning with Multiple GPT Agents

**Xiuyuan Hu**[1,2]*, **Guoqing Liu**[2]†, **Yang Zhao**[1], **Hao Zhang**[1]†
[1]Department of Electronic Engineering, Tsinghua University
[2]Microsoft Research AI4Science
huxy22@mails.tsinghua.edu.cn, guoqingliu@microsoft.com,
zhao-yang@tsinghua.edu.cn, haozhang@tsinghua.edu.cn

## Abstract

*De novo* drug design is a pivotal issue in pharmacology and a new area of focus in AI for science research. A central challenge in this field is to generate molecules with specific properties while also producing a wide range of diverse candidates. Although advanced technologies such as transformer models and reinforcement learning have been applied in drug design, their potential has not been fully realized. Therefore, we propose MolRL-MGPT, a reinforcement learning algorithm with multiple GPT agents for drug molecular generation. To promote molecular diversity, we encourage the agents to collaborate in searching for desirable molecules in diverse directions. Our algorithm has shown promising results on the GuacaMol benchmark and exhibits efficacy in designing inhibitors against SARS-CoV-2 protein targets. The codes are available at: `https://github.com/HXYfighter/MolRL-MGPT`.

## 1 Introduction

In recent years, significant strides have been made in computer-aided drug discovery (CADD) thanks to the development of various fields, including proteomics, genomics and deep learning [13, 66]. The conventional drug discovery process is typically time-consuming and financially demanding, with a low success rate. However, advanced machine learning techniques have the potential to reverse this predicament, greatly benefiting the economy and society's development [50, 58]. Currently, a major issue in the field of pharmacology is goal-directed *de novo* drug design, which involves generating new drug molecules with specific biochemical properties, such as designing compounds with high binding affinity to a designated protein target [15, 38]. Despite the numerous proposed machine learning algorithms for molecular generation, the chemical space is vast, and the relationship between molecular properties and structures is intricate, making it challenging to obtain satisfactory results in practical applications [42].

Molecular diversity is a critical concern in drug design because a diverse set of candidates can provide more choices for downstream screening and avoid drug resistance and unknown side effects [12, 41]. However, existing algorithms encounter challenges in designing diverse drug molecules. Many algorithms tend to generate sets of highly similar compounds, which is of little value for the subsequent drug development process.

Over the past several years, generative language models have made remarkable strides in natural language processing, vision, and audio. Among these models, the generative pre-trained transformer (GPT) is particularly notable, which has demonstrated impressive capabilities of language under-

---

*Work was done while Xiuyuan Hu was a research intern at Microsoft Research.
†Corresponding author: Hao Zhang (1st), Guoqing Liu.

37th Conference on Neural Information Processing Systems (NeurIPS 2023).

standing and generation [45, 46, 11]. In the field of chemistry, transformer-based language models pre-trained on the SMILES (simplified molecular input line entry system) representation of molecules have emerged. Through transfer learning, these models can be adapted to a range of tasks, including molecular property prediction, reaction prediction, and molecular optimization [26, 27, 4].

Reinforcement learning (RL) has emerged as a promising approach for *de novo* drug design [39, 68, 29]. The basic idea is to consider molecular property predictors (scoring functions) as rewards and train an RL agent to iteratively generate candidate compounds with increasingly high scores. These RL algorithms can explore the vast chemical space remarkably faster than human chemists, but they may have limitations in molecular diversity. Although multi-agent reinforcement learning (MARL) is a commonly used technique for promoting diversity in searching problems [34, 14], it has not yet been effectively utilized in the field of drug design.

To address the limitations of current approaches, we propose MolRL-MGPT (**Mol**ecular design using **R**einforcement **L**earning with **M**ultiple **GPT** agents), a novel MARL framework for *de novo* drug molecular design that utilizes GPT models as agents. Our approach treats molecular design as a cooperative Markov game, where multiple lightweight GPT agents collaborate to generate high-scoring molecules during the RL process. These agents share identical pre-trained parameters on a molecular SMILES dataset for initialization and have a common optimization objective for specific properties. To enhance the diversity of generated candidates, we incorporate an auxiliary loss function that encourages agents to explore in diverse directions. Our algorithm has demonstrated superior performance compared to various baselines on the GuacaMol benchmark. We also apply it to resolve the real-world problem of designing candidates against two SARS-CoV-2 protein targets, resulting in potentially desirable candidates with good binding affinity, drug-likeness, and synthetic accessibility. Moreover, we further validate the effectiveness of our design by comparative and ablation experiments on GNK3$\beta$, JNK3 and QED maximization tasks.

## 2   Related works

Machine learning has become a formidable instrument in molecular generation with applications to *de novo* drug design, aiding scientists in identifying novel molecules that possess the desired properties for drug discovery. The molecular representation is the basis of molecular generation and optimization algorithms, which can be roughly divided into three categories: 1D string, 2D image, and 3D geometry [15].

SMILES is the most commonly used 1D representation of molecules, which employs strings of characters to encode a molecule. Although the SMILES of each molecule is not unique, there is only one canonical SMILES, and each SMILES corresponds to a maximum of one molecule. It is noteworthy that most SMILES strings are invalid; that is, their corresponding structures cannot exist in the real world. Some algorithms and techniques applied in natural language processing (NLP) have been adopted to generate molecules using SMILES representations, such as variational autoencoder (VAE) [22, 16], recurrent neural network (RNN) [49], generative adversarial network (GAN) [23] and Bayesian optimization (BO) [37].

On the other hand, 2D and 3D molecular representations are more intuitive and have become popular in molecular design in recent years. For 2D molecular graphs with vertices corresponding to atoms and edges corresponding to chemical bonds between atoms, techniques such as graph neural network (GNN), genetic algorithm (GA) and flow network for graph data have already been applied to drug development [1, 28, 30, 62, 35, 21]. For 3D geometries of compounds, which theoretically contain the most structural information of the molecules, although models including diffusion have been introduced to the chemistry field [20, 33], they cannot currently design candidates with desired biochemical properties as 1D/2D *de novo* drug design approaches.

**RL-based drug design algorithms**   Reinforcement learning is the most popular technique for molecule generation. In molecular design, deep neural networks are usually employed as agents in studies that use RL for 1D string generation, which are then fine-tuned via customized reward functions [39, 9, 41, 59]. Likewise, when RL is applied to generating 2D molecular graphs, the states correspond to incomplete representations, and the actions involve adding substructures (such as atoms, bonds, and rings) to specific positions [65, 68, 29, 1, 63, 19]. What is more, recent studies

have incorporated 3D geometries into the RL process to consider the spatial properties of molecules during their generation [51, 52, 18].

**Transformers for chemical language**   Transformer is a type of deep neural network architecture entirely based on attention mechanism and has been widely applied in the field of NLP [57]. It has demonstrated a better ability to process long text sequences and parallel computing capabilities than traditional architectures such as RNN. Some works have already applied transformers in the field of chemistry. For instance, ChemFormer [26], MolGPT [4] and [24] have focused on SMILES-based pre-trained transformer models. Meanwhile, PROTAC-RL [67], TamGent [61] and [44] have concentrated on transformer-based drug design against protein targets, while MCMG [59] has used transformer as a component to enhance the learning capability of the algorithm.

**Diversity in drug design**   Previous research has proposed using reinforcement learning to generate diverse molecules for drug development [8, 59, 63, 41]. However, these studies have not adequately addressed this concern, and multi-agent reinforcement learning is yet to be effectively applied in *de novo* drug design.

## 3   Methodology

### 3.1   Problem definition

The fundamental aspect of the molecular generation problem formulation is a scoring function $s(x)$ of molecular properties, also known as an oracle, with the input of $x$ being a molecule and the output being a real number. Molecular properties can include physical properties such as molecular weight and the number of aromatic rings, as well as chemical properties such as logP and drug-likeness. Additionally, the molecular properties that we are most concerned with in real-world drug design are often related to biological activity, with binding affinity to protein targets being the most common, which can be estimated using docking software.

Standardly, the scoring function $s(\cdot)$ is typically constrained within the interval $[0, 1]$ when applied to a valid molecular input, where a higher score corresponds to a better molecular property. Invalid molecules consistently receive a score of -1. Researchers may implement a transformation function $t(\cdot)$ for each molecule property predictor $p(x)$ to achieve uniformity with the standard form, and create a multi-objective scoring function through a weighted combination of various oracles.

$$s(x) \in [0, 1] \cup \{-1\},$$
$$s(x) = t(p(x)) \text{ or } s(x) = \sum_i w_i \cdot t_i(p_i(x)), \ \sum_i w_i = 1 \tag{1}$$

The evaluation of a *de novo* molecule generation algorithm usually requires the assessment of a set of generated high-scoring molecules, and a common approach is to report the average score and diversity of the top-$k$ scoring generated molecules, where $k \in \mathbb{N}^+$ is given. A widely used metric of molecular diversity is internal diversity (IntDiv) [6]:

$$\text{IntDiv}(A) := \frac{1}{|A|(|A| - 1)} \sum_{(x,y) \in A \times A, x \neq y} d_T(\mathcal{F}(x), \mathcal{F}(y)) \tag{2}$$

where $A$ is a set of compounds, $d_T$ represents the Tanimoto distance [55], and $\mathcal{F}(x)$ refers to the extended-connectivity fingerprint (ECFP) [47] of a molecule $x$.

In drug molecular design tasks, we primarily focus on evaluating the final set of generated molecular candidates. Generally, we pay little attention to the generation process, including factors like time and computing resource consumption, as these costs are relatively insignificant compared to the conventional drug discovery process.

Hence, we can represent the problem of designing novel drug candidates as a cooperative Markov game consisting of multiple generative model agents. At every iteration $i$ $(i = 1, 2, \ldots, s)$, each agent $k$ $(k = 1, 2, \ldots, n)$ generates $m$ molecules (actions), and the scoring function acts as an environment providing rewards in the form of scores, which in turn are used to update the agents. The game's objective is primarily to maximize the average of the highest $k$ scores of generated molecules and secondarily to improve the diversity of the candidates.

## 3.2 MolRL-MGPT algorithm

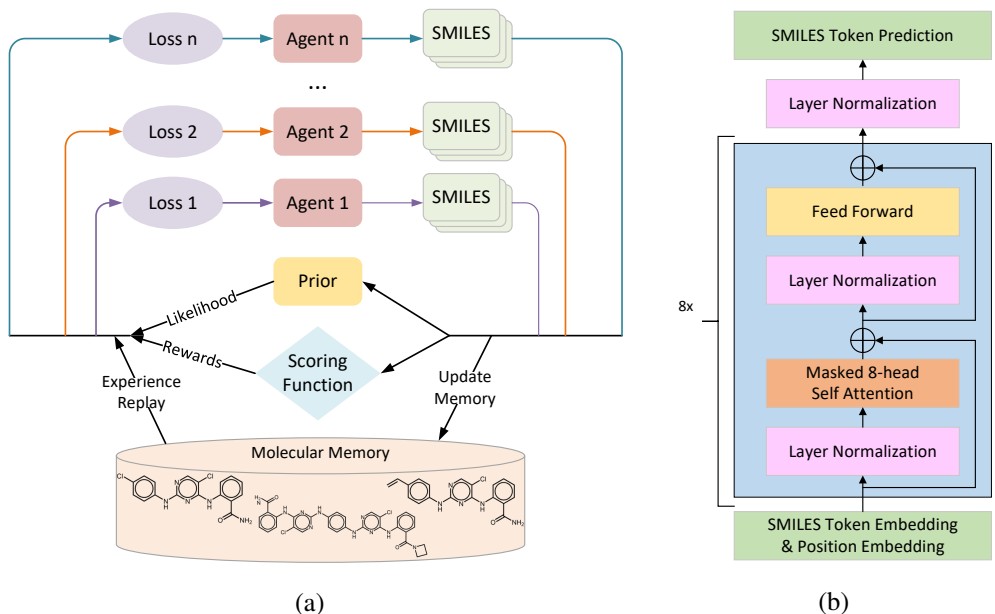

(a)               (b)

Figure 1: (a) Overview of our MolRL-MGPT algorithm. (b) The model architecture of the GPT prior model and agents.

As illustrated in Fig 1 (a), the MolRL-MGPT algorithm consists of iterations in which multiple agents are updated in a reinforcement learning way. Specifically, in each iteration, each agent first samples a set of SMILES strings with high likelihood and retrieves several high-scoring molecules in the past iterations from the molecular memory. Then, the scoring function is used to obtain the reward of each generated SMILES string. The GPT agents conduct this process in order. The loss functions are designed as follows.

**Loss function**  In MolRL-MGPT, the reward function is defined as the scores predicted by the scoring function. Our primary objective is to increase the average scores of SMILES strings generated by each agent. Furthermore, to prevent the agents from disregarding the knowledge acquired during the pre-training phase, we impose a penalty on the deviation between the new policies and the prior model, which is also employed to initialize all the agents. The loss function of the 1st GPT agent is designed as follows:

$$L_1(x; \Theta_1) = [\log P(x)_{\text{Prior}} - \log P(x)_{\text{Agent}_1} + \sigma_1 \cdot s(x)]^2 \quad (3)$$

where $\Theta_1$ is the parameters of the 1st agent, $x$ is a generated molecule, $\sigma_1$ is a coefficient for controlling the term of scores, and $P(x)_{\text{model}}$ refers to the likelihood of generating $x$ by model. It should be noted that typically $P(x)_{\text{Prior}} < P(x)_{\text{Agent}}$.

Additionally, we encourage the agents to explore different directions in the chemical space instead of conducting repetitive searches by introducing a term indicating the deviation between the current agent and previous ones. The loss function of the $k$th GPT agent is designed as follows:

$$\begin{aligned} L_k(x; \Theta_k) =& L_1(x; \Theta_k) - \sigma_2 \sum_{j=1}^{k-1} s(x) \cdot |\log P(x)_{\text{Agent}_k} - \log P(x)_{\text{Agent}_j}| \\ =& [\log P(x)_{\text{Prior}} - \log P(x)_{\text{Agent}_k} + \sigma_1 \cdot s(x)]^2 \\ & - \sigma_2 \sum_{j=1}^{k-1} s(x) \cdot |\log P(x)_{\text{Agent}t_k} - \log P(x)_{\text{Agent}_j}| \end{aligned} \quad (4)$$

where $k = 1, 2, \ldots n$, $\Theta_k$ is the parameters of the $k$th agent, and $\sigma_2$ is a coefficient for the encouragement term.

The selection of coefficients $\sigma_1, \sigma_2$ can be adapted according to the specific task at hand. Moreover, we propose to implement a decreasing schedule for $\sigma_1$ such that $\sigma_1$ decreases as $s(x)$ increases during the RL process. This is because we expect the increase in scores to correspond to a decrease in loss.

**Molecular Memory**    MolRL-MGPT utilizes a memory consisting of high-scoring molecules with a maximum size of 1000. The memory is updated with every SMILES string sampled, and compounds stored in the memory are sorted based on their scores.

### 3.3   Implementations

**Pre-training on chemical language**    Instead of the commonly used recurrent neural network (RNN) architecture for generating chemical language in SMILES strings, we adopt a transformer-based architecture. As shown in Fig 1 (b), our model is a mini version of the GPT model with 8 layers of transformer blocks, each consisting of 8 attention heads. The embedding size is 256, and the maximum length of SMILES strings is set to 128. The pre-training approach of our model involved learning molecular structure patterns. Our prior model has only around 6.4M parameters, much less than the GPT-2 model [46].

Two versions of the prior model are trained using unsupervised learning on the ChEMBL [36] and ZINC-100M [53] datasets, respectively. The data are preprocessed by removing molecules with ionized structures and SMILES strings longer than 100 characters, which are not typical for small molecule drugs. Additionally, we use SMILES randomization for data augmentation, which has been proven helpful in enhancing generating capacity [3]. Ten training epochs are conducted with a batch size of 2048 and a maximum learning rate of 0.001, using a learning rate schedule featuring warm-up and cosine decay. The training on ChEMBL (roughly 2 million SMILES) takes around 5 hours using a single NVIDIA A100 GPU.

The pre-trained SMILES models should be objectively evaluated by the valid ratio of the generated molecules, representing the proportion of valid molecules present in the generated SMILES. Generally, this value can exceed 90% [43]. Our pre-trained transformer model has achieved a valid ratio of 98% in generating molecules, indicating its success in capturing the inherent grammar and rules of molecular SMILES strings.

**Experience replay**    Experience replay is a widely employed technique in deep reinforcement learning. The agent or learner records its experience in a memory buffer and randomly samples past experiences, enabling it to learn from previous experiences with reduced correlation between consecutive training samples. This technique helps agents avoid overfitting, generalize to unseen situations, and achieve a more stable and efficient learning process [32]. In MolRL-MGPT, we replay the "successful" experiences by randomly sampling 5 molecules from the 25 highest-scoring molecules in the molecular memory and computing the loss on these molecules together with SMILES strings generated in the current iteration.

**\*Similarity penalization**    Some previous works for *de novo* drug design add a penalizing term for high similarity of identical skeletons between new molecules and previously found high-scoring molecules in order to encourage the agent to search in unexplored space rather than repeatedly finding the same or similar compounds [9, 60]. However, our experiments indicate that this trick does not work with our algorithm.

## 4   Experiments

To demonstrate the performance of MolRL-MGPT, we conduct three groups of experiments. Firstly, we run our algorithm on the public *de novo* molecular design benchmark, GuacaMol, and compare it with existing advanced methods. Secondly, we apply MolRL-MGPT to the design of inhibitors against SARS-CoV-2 protein targets, which is a current real-world challenge for human beings. Thirdly, we conduct comparative and ablation experiments on GNK3$\beta$, JNK3 and QED maximization tasks to validate the effectiveness of modules in our design.

## 4.1 GuacaMol benchmark

GuacaMol [10] is a widely recognized open-source evaluation framework for *de novo* molecular design algorithms. It contains 20 goal-directed tasks mimicking the drug discovery objectives, corresponding to 20 standard scoring functions described in Sec 3.1. These tasks cover commonly used objectives in drug design, such as structural, physicochemical, and biochemical properties.

For comparison, we select the following baselines: (1) SMILES GA [64], a genetic algorithm based on SMILES; (2) SMILES LSTM [49], an LSTM network generating SMILES strings autoregressively, combined with a hill-climb algorithm for optimization; (3) Graph GA [28], a genetic algorithm with crossovers and mutations performed on molecular graphs; (4) Reinvent [39], a deep reinforcement learning framework for training an RNN model generating SMILES; (5) GEGL [1], a genetic expert-guided learning framework for training an RNN for molecular generation. Results of some other baselines are shown in the Appendix.

The prior model for MolRL-MGPT was pre-trained on the official GuacaMol dataset, which is a subset of ChEMBL. The hyper-parameters are set such that we run each tasks for 5000 tasks (break if the score has achieved 1.000) with 4 GPT agents, and the batch size (number of sampled SMILES strings) of each agent is 256. The values of coefficients are set to $\sigma_1 = 1000$ with a linear decreasing schedule, and $\sigma_2 = 0.1$. The entire set of 20 tasks tasks less than 400 hours to complete when run on a single NVIDIA A100 GPU.

Table 1: Scores of MolRL-MGPT and other baselines on the GuacaMol benchmark.

| Tasks | SMILES GA | SMILES LSTM | Graph GA | Reinvent | GEGL | MolRL-MGPT |
|---|---|---|---|---|---|---|
| 1. Celecoxib rediscovery | 0.732 | **1.000** | **1.000** | **1.000** | **1.000** | **1.000** |
| 2. Troglitazone rediscovery | 0.515 | **1.000** | **1.000** | **1.000** | 0.552 | **1.000** |
| 3. Thiothixene rediscovery | 0.598 | **1.000** | **1.000** | **1.000** | **1.000** | **1.000** |
| 4. Aripiprazole similarity | 0.834 | **1.000** | **1.000** | **1.000** | **1.000** | **1.000** |
| 5. Albuterol similarity | 0.907 | **1.000** | **1.000** | **1.000** | **1.000** | **1.000** |
| 6. Mestranol similarity | 0.790 | **1.000** | **1.000** | **1.000** | **1.000** | **1.000** |
| 7. $C_{11}H_{24}$ | 0.829 | 0.993 | 0.971 | 0.999 | **1.000** | **1.000** |
| 8. $C_9H_{10}N_2O_2PF_2Cl$ | 0.889 | 0.879 | 0.982 | 0.877 | **1.000** | 0.939 |
| 9. Median molecules 1 | 0.334 | 0.438 | 0.406 | 0.434 | **0.455** | 0.449 |
| 10. Median molecules 2 | 0.380 | 0.422 | 0.432 | 0.395 | **0.437** | 0.422 |
| 11. Osimertinib MPO | 0.886 | 0.907 | 0.953 | 0.889 | **1.000** | 0.977 |
| 12. Fexofenadine MPO | 0.931 | 0.959 | 0.998 | **1.000** | **1.000** | **1.000** |
| 13. Ranolazine MPO | 0.881 | 0.855 | 0.920 | 0.895 | 0.933 | **0.939** |
| 14. Perindopril MPO | 0.661 | 0.808 | 0.792 | 0.764 | **0.833** | 0.810 |
| 15. Amlodipine MPO | 0.722 | 0.894 | 0.894 | 0.888 | 0.905 | **0.906** |
| 16. Sitagliptin MPO | 0.689 | 0.545 | **0.891** | 0.539 | 0.749 | 0.823 |
| 17. Zaleplon MPO | 0.413 | 0.669 | 0.754 | 0.590 | 0.763 | **0.790** |
| 18. Valsartan SMARTS | 0.552 | 0.978 | 0.990 | 0.095 | **1.000** | 0.997 |
| 19. deco hop | 0.970 | 0.996 | **1.000** | 0.994 | **1.000** | **1.000** |
| 20. scaffold hop | 0.885 | 0.998 | **1.000** | 0.990 | **1.000** | **1.000** |
| Total | 14.396 | 17.340 | 17.983 | 16.350 | 17.627 | **18.052** |

As shown in Table 1, the MolRL-MGPT algorithm outperforms baselines in 13 molecular design tasks in the GuacaMol benchmark, and its total score of 20 tasks also ranks first. These results indicate that MolRL-MGPT excels in general scenarios of *de novo* drug design.

## 4.2 Designing inhibitors against SARS-CoV-2 targets

Molecular docking is a computational method used to predict the binding modes of small molecules to a protein target. It involves predicting the spatial orientation and binding affinity of the small molecule in the active site of the protein. This information is useful in drug discovery since it enables identifying potential drug candidates and understanding how they interact with their targets. Autodock Vina [56] is currently the most widely used molecular docking software. However, we opt to use Quick Vina 2 [2], a novel and more efficient alternative.

The quantitative binding affinity is named docking score, which is calculated based on the energies of the interaction between the ligand and the receptor, and a lower docking score indicates a more stable and, therefore, more likely binding pose. Typically, docking scores are negative, and desirable docking scores range from -10 to -14 kcal/mol. Therefore, we use a reverse sigmoid function as the transformation function of the docking score:

$$t_{\text{docking}}(p) = \frac{1}{1 + 10^{0.625 \cdot (p+10)}} \tag{5}$$

SARS-CoV-2 (Severe Acute Respiratory Syndrome Coronavirus 2), commonly referred to as the novel coronavirus, is a respiratory virus which, in late 2019, emerged as a global pandemic leading to the COVID-19 disease that mainly targets the respiratory system. This disease is a severe public health concern that continues to pose a crisis worldwide and requires a comprehensive response to mitigate its spread and negative effects. Therefore, we apply our algorithm to the design of inhibitors against protein targets of SARS-CoV-2, which is a significant real-world issue. Following [48], we select two targets: `PLPro_7JIR`[3] and `RdRp_6YYT`[4].

Besides docking scores, we also consider two additional oracles often employed in practical drug design: (1) **QED** (Quantitative Estimate of Drug-likeness), which quantifies the drug-likeness of a molecule based on the concept of desirability of eight molecular properties [7], ranging in $[0, 1]$; (2) **SA** (Synthetic Accessibility), which incorporates fragment contributions and a complexity penalty [17], ranging in $[1, 10]$. Their transformation functions are linear:

$$t_{\text{QED}}(p) = p, \quad t_{\text{SA}}(p) = \frac{10 - p}{9} \tag{6}$$

Moreover, the scoring function for designing inhibitors against SARS-CoV-2 protein targets is a linear combination of docking scores, QED scores and SA scores:

$$s_{\text{total}}(x) = 0.8 \cdot s_{\text{docking}}(x) + 0.1 \cdot s_{\text{QED}}(x) + 0.1 \cdot s_{\text{SA}}(x) \tag{7}$$

We run the RL process for 1000 iterations on each target with 4 GPT agents, and the batch size of each agent is 128. The whole process for one target takes approximately 100 hours on a single NVIDIA A100 GPU and 64 CPU cores.[5] Details of the generated candidates are as follows.

**PLPro_7JIR target** PLPro (papain-like protease) is an attractive target for SARS-CoV-2 since it plays a fundamental role in cleavage and maturation of viral polyproteins, assembly of the replicase-transcriptase complex, and disruption of host responses. 7JIR is a C111S mutant form of the structure of PLPro [40]. Three candidate inhibitors against the PLPro_7JIR target generated by MolRL-MGPT are shown in Table 2.

Table 2: Candidate inhibitors against the PLPro_7JIR target generated by MolRL-MGPT. The SMILES of the three candidates are:
(1) O=C(c1cccc(-c2ccc3c(-c4nc5cc(F)c(F)cc5[nH]4)n[nH]c3c2)c1)c1ccccc1;
(2) c1(-c2cc(-c3cc4c(cc3)snn4)ccc2)ccc(C(NCc2cc(C(=O)Nc3nn[nH]n3)ccc2F)=O)cc1;
(3) c1c(-c2c(C)ccc(C(=O)c3cc4ccccc4[nH]3)c2)nc(-c2ccnc(-c3cccc(-c4ccccc4)c3)c2)cc1.

| Molecule | | | |
|---|---|---|---|
|  |  |  |
| docking score (↓) | -11.3 | -11.1 | -11.2 |
| QED score (↑) | 0.310 | 0.258 | 0.214 |
| SA score (↓) | 2.530 | 2.729 | 2.549 |

---

[3] https://www.rcsb.org/structure/7JIR

[4] https://www.rcsb.org/structure/6YYT

[5] Docking consumes more than 95% of the time, although parallel computing of Quick Vina software is implemented.

**RdRp_6YYT target**  RdRp (RNA-dependent RNA polymerase) works for the replication of genome and the transcription of genes of SARS-CoV-2, and 6YYT is the PDB identification code of its structure [25]. Three candidate inhibitors against the RdRp_6YYT target generated by MolRL-MGPT are shown in Table 3.

Table 3: Candidate inhibitors against the RdRp_6YYT target generated by MolRL-MGPT. The SMILES of the three candidates are:
(1) O=C1CCN(c2nc(-c3cccc(C(=O)N4CCN(c5cc(-c6ccccc6)nc6ccccc56)CC4)c3)cc3ccccc23)CCN1;
(2) O=C1CCN(c2nc(-c3cccc(C(=O)N4CCN(c5cc(-c6nc7ccccc7c(=O)[nH]6)c6ccccc6n5)CC4)c3)nc3ccccc23)CCN1;
(3)  Cc1ccc(-c2cc(N3CCN(C(=O)c4cccc(-c5nc(N6CCNC(=O)CC6)c6ccccc6n5)c4)CC3)c3ccccc3n2)cc1.

| Molecule | | | |
|---|---|---|---|
| docking score ($\downarrow$) | -12.3 | -13.1 | -13.2 |
| QED score ($\uparrow$) | 0.237 | 0.253 | 0.241 |
| SA score ($\downarrow$) | 2.772 | 3.104 | 2.806 |

The generated candidates against the PLPro_7JIR and RdRp_6YYT targets exhibit desirable binding affinities (docking scores) and synthetic accessibility. Although the drug-likeness scores of these candidates may not be high, it is reasonable since QED is estimated based on the distribution of existing drug molecules. All these drugs are ineffective in inhibiting the two targets of SARS-CoV-2; thus, designing new dissimilar compounds becomes necessary.

## 4.3  GSK3$\beta$, JNK3 and QED maximization

To demonstrate the effectiveness of our design, we perform ablation experiments on commonly used oracles for simulating real-world drug design with low consumption: GSK3$\beta$ and JNK3. Their scores are estimated by random forests trained on ExCAPE-DB dataset [54], which measure the bioactivities of molecules against the Glycogen synthase kinase 3 beta target (GSK3$\beta$) and the c-Jun N-terminal kinase 3 target (JNK3). Previous research has shown that inhibiting these targets can benefit the treatment of Alzheimer's Disease [31].

We carry out ablation experiments to validate the following settings of MolRL-MGPT:

1. The optimal number of agents for a fixed total batch size,
2. Whether the loss term that encourages agents to search in different directions (ED) really works,
3. Whether the experience replay (ER) really works,
4. Whether the decreasing schedule of $\sigma_1$ (DS) really works,
5. Whether the possible technique of similarity penalization (SP) works [6].

The base algorithm is denoted as MolRL-MGPT, which consists of 4 agents and incorporates the modules ED, ER, DS, without SP. For both the GSK3$\beta$ and JNK3 tasks, we present the average and standard deviation of the mean scores and internal diversities of the top-100 high-scoring molecules generated by each strategy.

Besides ablation experiments, we also compare the performance of MolRL-MGPT with several baselines on GSK3$\beta$, JNK3 and QED maximization tasks. The baselines are Graph GA [28], Reinvent [39], JT-VAE [30] and GFlowNet [5]. More details are provided in the Appendix.

Table 4 indicates that the algorithm demonstrates the best performance with four agents, and additional agents do not improve the results. Encouraging agents to explore different paths in the chemical

---

[6]Penalize on the score of a molecule if its similarity to one of previously generated candidates is larger than 0.8.

Table 4: Results of experiments on GSK3$\beta$ and JNK3 maximization.

| | GSK3$\beta$ top-100 | | JNK3 top-100 | |
| --- | --- | --- | --- | --- |
| | mean score | IntDiv | mean score | IntDiv |
| 1 agent | 1.000 ± 0.000 | 0.318 ± 0.020 | 0.954 ± 0.012 | 0.343 ± 0.017 |
| 2 agents | 1.000 ± 0.000 | 0.335 ± 0.017 | 0.960 ± 0.012 | 0.357 ± 0.028 |
| **MolRL-MGPT** | 1.000 ± 0.000 | 0.362 ± 0.015 | 0.961 ± 0.010 | 0.372 ± 0.025 |
| 8 agents | 1.000 ± 0.000 | 0.360 ± 0.020 | 0.958 ± 0.015 | 0.369 ± 0.018 |
| w/o ED | 1.000 ± 0.000 | 0.285 ± 0.023 | 0.961 ± 0.008 | 0.345 ± 0.025 |
| w/o ER | 0.964 ± 0.005 | 0.332 ± 0.019 | 0.918 ± 0.008 | 0.356 ± 0.023 |
| w/o DS | 0.997 ± 0.001 | 0.358 ± 0.016 | 0.940 ± 0.014 | 0.370 ± 0.027 |
| w/ SP | 1.000 ± 0.000 | 0.360 ± 0.021 | 0.956 ± 0.009 | 0.365 ± 0.015 |
| GFlowNet | 0.649 ± 0.072 | 0.715 ± 0.104 | 0.437 ± 0.219 | 0.716 ± 0.145 |
| GraphGA | 0.919 ± 0.016 | 0.365 ± 0.024 | 0.875 ± 0.025 | 0.380 ± 0.015 |
| JT-VAE | 0.235 ± 0.083 | 0.770 ± 0.067 | 0.159 ± 0.040 | 0.781 ± 0.127 |
| Reinvent | 0.965 ± 0.011 | 0.308 ± 0.035 | 0.942 ± 0.019 | 0.368 ± 0.021 |

Table 5: Results of experiments on QED maximization.

| | QED top-100 | |
| --- | --- | --- |
| | mean score | IntDiv |
| **MolRL-MGPT** | 0.948 ± 0.000 | 0.862 ± 0.004 |
| GFlowNet | 0.938 ± 0.001 | 0.809 ± 0.017 |
| GraphGA | 0.928 ± 0.001 | 0.845 ± 0.005 |
| JT-VAE | 0.921 ± 0.003 | 0.856 ± 0.012 |
| Reinvent | 0.948 ± 0.000 | 0.658 ± 0.035 |

space does promote diversity of the generated molecules, and using experience replay and decreasing schedule of $\sigma_1$ benefits mean scores of candidates. Additionally, the similarity penalization trick does not appear to be effective in MolRL-MGPT. As a consequence, we have verified the effectiveness of our design.

As shown in Table 4 and 5, compared with baselines with competitive mean scores, MolRL-MGPT performs better in internal diversity.

## 5 Conclusion and discussion

In this paper, we present MolRL-MGPT, a multi-agent reinforcement learning framework for *de novo* drug molecular design, which adopts transformer models as agents and the fundamental idea is to encourage agents to collaborate to search with different directions in the chemical space. MolRL-MGPT demonstrates superior performance on the GuacaMol benchmark and does well in designing inhibitors against SARS-CoV-2 protein targets.

Admittedly, there exist some possible approaches for further improvements to our algorithm:

- **Better data source**: As we all know, data sources play a decisive role in the performance of ML algorithms in chemistry and biology. Higher quality pre-training data or more annotated data tailored to specific tasks may further improve the performance of MolRL-MGPT.

- **Scoring function**: The design of the scoring function may also improve the algorithm's performance. For example, in multi-property joint tasks, adjusting the coefficients of terms in the scoring function may be beneficial, and more accurate and fast software for predicting molecular properties is also helpful.

- **Insights for specific objectives**: In practical drug development, with more specialized and in-depth research on each objective, we should utilize the knowledge specific to certain tasks more fully to design candidate drug molecules better.

In summary, MolRL-MGPT is a promising and feasible approach for *de novo* drug design. It provides pharmaceutical researchers with a fast and effective method to generate a diverse set of molecular structures that meet specified conditions, as long as the scoring functions of these conditions are provided. We believe that MolRL-MGPT will be of great assistance in drug discovery.

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

# A    GuacaMol benchmark

Table 6: More results of the experiments on the GuacaMol benchmark.

| Tasks | dataset | Graph MCTS | GFlowNet | MolRL-MGPT |
|---|---|---|---|---|
| 1. Celecoxib rediscovery | 0.505 | 0.355 | 0.409 | $\mathbf{1.000} \pm 0.000$ |
| 2. Troglitazone rediscovery | 0.419 | 0.311 | 0.211 | $\mathbf{1.000} \pm 0.000$ |
| 3. Thiothixene rediscovery | 0.456 | 0.311 | 0.342 | $\mathbf{1.000} \pm 0.000$ |
| 4. Aripiprazole similarity | 0.595 | 0.380 | 0.586 | $\mathbf{1.000} \pm 0.000$ |
| 5. Albuterol similarity | 0.719 | 0.749 | 0.458 | $\mathbf{1.000} \pm 0.000$ |
| 6. Mestranol similarity | 0.629 | 0.402 | 0.396 | $\mathbf{1.000} \pm 0.000$ |
| 7. $C_{11}H_{24}$ | 0.684 | 0.410 | 0.535 | $\mathbf{1.000} \pm 0.000$ |
| 8. $C_9H_{10}N_2O_2PF_2Cl$ | 0.747 | 0.631 | 0.224 | $0.939 \pm 0.003$ |
| 9. Median molecules 1 | 0.334 | 0.225 | 0.218 | $0.447 \pm 0.006$ |
| 10. Median molecules 2 | 0.351 | 0.170 | 0.195 | $0.423 \pm 0.004$ |
| 11. Osimertinib MPO | 0.839 | 0.784 | 0.792 | $0.977 \pm 0.001$ |
| 12. Fexofenadine MPO | 0.817 | 0.695 | 0.715 | $\mathbf{1.000} \pm 0.000$ |
| 13. Ranolazine MPO | 0.792 | 0.616 | 0.680 | $\mathbf{0.939} \pm 0.000$ |
| 14. Perindopril MPO | 0.575 | 0.385 | 0.459 | $0.809 \pm 0.005$ |
| 15. Amlodipine MPO | 0.696 | 0.533 | 0.430 | $\mathbf{0.906} \pm 0.001$ |
| 16. Sitagliptin MPO | 0.509 | 0.458 | 0.042 | $0.822 \pm 0.003$ |
| 17. Zaleplon MPO | 0.547 | 0.488 | 0.072 | $\mathbf{0.790} \pm 0.008$ |
| 18. Valsartan SMARTS | 0.259 | 0.040 | 0.000 | $0.997 \pm 0.000$ |
| 19. deco hop | 0.933 | 0.590 | 0.587 | $\mathbf{1.000} \pm 0.000$ |
| 20. scaffold hop | 0.738 | 0.478 | 0.475 | $\mathbf{1.000} \pm 0.000$ |
| Total | 12.144 | 9.009 | 7.826 | $\mathbf{18.049} \pm 0.003$ |

# B    Designing inhibitors against SARS-CoV-2 targets

Table 7: The mean scores and internal diversity of the top-100 drug candidates against the PLPro_7JIR target generated by MolRL-MGPT and other baselines.

| Methods | Docking score ($\downarrow$) | QED score ($\uparrow$) | SA score ($\downarrow$) | IntDiv |
|---|---|---|---|---|
| JT-VAE | -8.76 ± 0.35 | 0.795 ± 0.038 | 2.994 ± 0.140 | 0.836 ± 0.032 |
| GFlowNet | -9.11 ± 0.21 | 0.726 ± 0.015 | 2.823 ± 0.076 | 0.825 ± 0.010 |
| GraphGA | -10.83 ± 0.08 | 0.380 ± 0.013 | 3.638 ± 0.162 | 0.740 ± 0.017 |
| Reinvent | -10.75 ± 0.05 | 0.392 ± 0.008 | 2.649 ± 0.035 | 0.619 ± 0.023 |
| **MolRL-MGPT** | **-11.02 ± 0.06** | **0.386 ± 0.006** | **2.550 ± 0.047** | **0.745 ± 0.008** |

Table 8: The mean scores and internal diversity of the top-100 drug candidates against the RdRp_6YYT target generated by MolRL-MGPT and other baselines.

| Methods | Docking score ($\downarrow$) | QED score ($\uparrow$) | SA score ($\downarrow$) | IntDiv |
|---|---|---|---|---|
| JT-VAE | -8.33 ± 0.25 | 0.719 ± 0.019 | 2.959 ± 0.094 | 0.828 ± 0.018 |
| GFlowNet | -8.89 ± 0.16 | 0.656 ± 0.033 | 2.854 ± 0.061 | 0.770 ± 0.015 |
| GraphGA | -11.26 ± 0.12 | 0.262 ± 0.010 | 3.520 ± 0.049 | 0.658 ± 0.009 |
| Reinvent | -11.30 ± 0.04 | 0.275 ± 0.006 | 2.917 ± 0.035 | 0.616 ± 0.021 |
| **MolRL-MGPT** | **-11.84 ± 0.07** | **0.278 ± 0.005** | **2.894 ± 0.072** | **0.670 ± 0.013** |

