# OpenReview forum: "De novo Drug Design using Reinforcement Learning with Multiple GPT Agents"
_NeurIPS.cc/2023/Conference — NeurIPS 2023 poster_

### Official Review · Reviewer_AzpA · 2023-07-02

**Soundness:** 4 excellent
**Presentation:** 4 excellent
**Contribution:** 2 fair
**Rating:** 6
**Confidence:** 3

**Summary:**

This paper proposes a method named MolRL-MGPT for drug molecular generation.
Concretely, GPT-based agents are used to iteratively generate candidate compounds, and a special reward signal is adpoted to encourage agents to explore in diverse directions.
The experiments on GuacaMol benchmark show the superiority of the method.


**Strengths:**

1.	An effective method for De novo Drug Design is proposed.
2.	A special reward signal is proposed to promote the diversity of the agents.


**Weaknesses:**

It is not appropriate to call the proposed method a “RL-based” method though the score function can be treated as the “reward”. RL follows Markov decision process and aims to maximize the accumulated return, but it seems that the goal in this paper is to maximize the final score. Meanwhile, there is no RL objective function in this paper. (I suggest to change the RL-related statement to another, and this will not affect my rating.)
1.	Eq.3 aims to make each agent get a fix improvement which neglect the difficulty of different generation timesteps.
2.	Eq.4 forcing the different agents obtain different scores, but different score is not equal to the different explore direction.
3.	It seems that the agents are treated as independent units without sharing their experience. Since the different agents cooperate with each other and aim to achieve a common goal, different kinds of correlation need to be considered.


**Questions:**

1.	Though there are some transformer-based models for chemical language. There still exists a gap between molecules structure and natural language. How the model will perform when the GPT is replaced with graph network?
2.	How to determine the suitable hyperparameters in real-world applications?


**Limitations:**

No potential negative societal impact.

---

> ### Author Rebuttal · Authors · 2023-08-07
>
> Thanks for your valuable review comments! We address your main concerns below:
>
> **Q1 (in *weaknesses*)**: It is not appropriate to call the proposed method a “RL-based” method though the score function can be treated as the “reward”. RL follows Markov decision process and aims to maximize the accumulated return, but it seems that the goal in this paper is to maximize the final score. Meanwhile, there is no RL objective function in this paper.
>
> **A1**: I concur with your opinion that our algorithm diverges somewhat from the conventional understanding of "reinforcement learning". However, we employ this terminology primarily because, in the context of molecular design for computer-aided drug discovery, "reinforcement learning" serves as the widely recognized term for denoting a category of multi-step generation algorithms featuring reward functions (https://www.sciencedirect.com/science/article/pii/S0928098722002093), and our algorithm falls within the scope of this methodological category. Moreover, in section 2 (related works), we allocate a paragraph to provide an overview of "RL-based drug design algorithms", which contains various previous works in this category.
>
> **Q2 (in *weaknesses*)**: 1. Eq.3 aims to make each agent get a fix improvement which neglect the difficulty of different generation timesteps. 2. Eq.4 forcing the different agents obtain different scores, but different score is not equal to the different explore direction. 3. It seems that the agents are treated as independent units without sharing their experience. Since the different agents cooperate with each other and aim to achieve a common goal, different kinds of correlation need to be considered.
>
> **A2**: It seems that you are a little confused about our algorithm design, so I will clarify certain aspects of our design in further detail.
>
> 1. Eq.3 is the loss function of the first agent in the RL process. Its purpose is to encourage the agent to learn the characteristics of high-scoring molecules as far as possible without deviating too much from the prior parameter. At each RL step, each term of the loss function is recalculated, so the improvement of the agent is not fixed. In addition, during the RL process, it does tend to be progressively more challenging for the agent to improve the molecular score, but this does not contradict the objective of the loss function to improve the molecular score.
> 2. Eq.4 is not intended to force different agents to generate molecules with different scores, but rather to motivate all agents to comprehend the characteristics of high-scoring molecules while simultaneously exploring the chemical space in diverse directions. We accomplish the second objective by rewarding the difference in the probability of generating the same SMILES by different agents, which is similar to the idea in Eq.3 of guaranteeing the effectiveness of agents by punishing the difference in the probability of generating the same SMILES for agents and the prior model.
> 3. Indeed, we do not treat agents as independent units in our algorithm. This is evident in Eq.4 where we explicitly reward the differences between agents. You also mention some ideas in the design of cooperative multi-agent reinforcement learning algorithms, but they might not be directly applicable to the specific framework of our algorithm design.
>
> **Q3 (in *questions*)**: Though there are some transformer-based models for chemical language, there still exists a gap between molecules structure and natural language. How the model will perform when the GPT is replaced with graph network?
>
> **A3**: This question poses an important inquiry into the effectiveness of distinct molecular representations.  Notably, the Simplified Molecular Input Line Entry System (SMILES) string we employ encapsulates identical structural information as the two-dimensional molecular graph. Hence, the potential of SMILES-based molecular design algorithms parallels that of their 2D molecular graph counterparts.
>
> Furthermore, recent research has demonstrated the superior performance of SMILES-based reinforcement learning algorithms compared to graph-based models in molecular design tasks (https://openreview.net/forum?id=yCZRdI0Y7G). This outcome underscores our confidence that our algorithm using GPT to generate SMILES strings competes robustly with graph-based approaches.
>
> It is worth acknowledging that both SMILES strings and molecular graphs inherently sacrifice spatial information in comparison to actual three-dimensional molecules. Nonetheless, the existing algorithms for generating *de novo* molecules in 3D have not reached a stage where they can supplant the effectiveness of string-based and graph-based methods. I hold the view that should a significantly advanced 3D generation model emerge, its integration to replace the GPT agent in our algorithm would hold substantial promise as a future direction.
>
> **Q4 (in *questions*)**: How to determine the suitable hyperparameters in real-world applications?
>
> **A4**: In the RL process of MolRL-MGPT, the pivotal hyper-parameters include: the number of agents, the number of RL steps, learning rate, $\sigma_1$, $\sigma_2$, the number of samples for experience replay, etc. Among them, our experiments in section 4.3 suggest that 4 is the best number of agents. Regarding the number of RL steps, it's advisable to extend this count as far as practicable, until the agents show no significant progress for many steps. Other hyper-parameters do need to be tuned specifically in each design task (by monitoring the changing curves of the agents throughout the RL process), where the learning rate and $\sigma_2$ are particularly sensitive. For our experiments, the setting of hyper-parameters can be found in the paper and codes.

---

> > ### Comment · Reviewer_AzpA · 2023-08-21
> >
> > Thanks for your reply, but some of the concerns still exists.
> >
> > 1. The first 2 terms of Eq.3 regularize the parameter deviating. However,  the 3 term seems can not "encourage the agent to learn the characteristics of high-scoring molecules as far as possible", since the score function may be inaccurate and the first 2 terms is also unstable, especially at the beginning of the training. Therefore, I doubt the effectiveness of Eq. 3. It looks too strange and lack the theoretical guarantees. Please refer to PPO[1] for a possible way.
> >
> > 2. From the rebuttal, the 1st term of Eq. 4 " motivate all agents to comprehend the characteristics of high-scoring molecules". However, it is not related  to explore "the chemical space in diverse directions", and I can not understand where the "diversity" comes. Exploration in RL means choose a non-optimal action to prevent the model to fall into sub-optimal results, and the "exploration"  in Eq. 4 seems can not try 'non-optimal action' but only encourage high score. We have no evidence that different molecules with "multiple high scores" is  equivalent to "diversity" since they may be similar.
> >
> > 3.  "Explicitly reward the differences between agents" is not equal to build the relationships among agents since the rewards only contain very little information.
> >
> > 4. Since large language model (LLM) is trained via natural language, the effectiveness of LLM in molecules still needs further evidence.
> >
> >
> >
> > [1] Proximal Policy Optimization Algorithms

---

> > > ### Author Response · Authors · 2023-08-21
> > >
> > > Thank you for your reply! Our answers to your new questions are as follows:
> > >
> > > > The first 2 terms of Eq.3 regularize the parameter deviating. However, the 3 term seems can not "encourage the agent to learn the characteristics of high-scoring molecules as far as possible", since the score function may be inaccurate and the first 2 terms is also unstable, especially at the beginning of the training. Therefore, I doubt the effectiveness of Eq. 3. It looks too strange and lack the theoretical guarantees. Please refer to PPO[1] for a possible way.
> > >
> > > Our design of Eq.3 is similar to the RL loss function of Reinvent (https://jcheminf.biomedcentral.com/articles/10.1186/s13321-017-0235-x) and is indeed innovative compared to the original PPO. It is true that the scoring function may be inaccurate, but the RL training is typically stable as long as well-trained models and reasonable hyper-parameters are used. You can verify the validity of our design by running our code.
> > >
> > > > From the rebuttal, the 1st term of Eq. 4 " motivate all agents to comprehend the characteristics of high-scoring molecules". However, it is not related to explore "the chemical space in diverse directions", and I can not understand where the "diversity" comes. Exploration in RL means choose a non-optimal action to prevent the model to fall into sub-optimal results, and the "exploration" in Eq. 4 seems can not try 'non-optimal action' but only encourage high score. We have no evidence that different molecules with "multiple high scores" is equivalent to "diversity" since they may be similar.
> > >
> > > The "diversity" mainly comes from the 2nd term of Eq.4, which encourages agents to search for diverse molecules. Moreover, to be clear, our algorithm does not use the exploration-exploitation paradigm, and by "exploration" we mean "searching".
> > >
> > > > "Explicitly reward the differences between agents" is not equal to build the relationships among agents since the rewards only contain very little information.
> > >
> > > You are right, but our objective is not to make agents share their full experience, just to avoid them falling into the same local optima. Our design is sufficient for this purpose.
> > >
> > > > Since large language model (LLM) is trained via natural language, the effectiveness of LLM in molecules still needs further evidence.
> > >
> > > Our agents use the GPT architecture, but they are pre-trained on a dataset of chemical molecules (SMILES strings), not natural language. In addition, we have presented previous related works in the original paper (line 92), including ChemFormer (https://iopscience.iop.org/article/10.1088/2632-2153/ac3ffb) and MolGPT (https://pubs.acs.org/doi/10.1021/acs.jcim.1c00600), which can demonstrate that LLMs are also effective for chemical language. Our work aims to further explore the potential of LLMs for chemical language.

---

### Official Review · Reviewer_4Wce · 2023-07-05

**Soundness:** 3 good
**Presentation:** 3 good
**Contribution:** 3 good
**Rating:** 7
**Confidence:** 4

**Summary:**

This paper creates a multi-agent reinforcement learning approach to promote diversity in the search space of small molecules during molecular optimization.  Because of the complicated nature of early stage drug discovery research, the diversity is useful during early work in the identification and validation of small molecules, however previous RL methods did not manage to address diversity in a satisfactory way.  The algorithm is rather simple and employs pre-trained GPT agents that speak the language of molecular SMILES. Importantly, the algorithm manages to outperform previous state of the art methods on a number of public benchmarks.


**Strengths:**

The use of multiple agents in this small molecule optimization setting is original and proves to be very useful.  The explicit diversification by equation (4) ends up creating a stronger model than previous attempts that used RL or other deep learning methodologies, and importantly, it outperforms (even if only barely) the graph Genetic Algorithm, a significantly simpler algorithm that puts other methods to shame in several benchmarks.  The paper demonstrates the increase in diversity of the designs with the increasing number of agents.  The paper is written in a clear fashion, and the main result it significant.  The authors provide code.


**Weaknesses:**

A minor weakness of this method is that the performance of the code is somewhat slow, which is somewhat understandable given the slow individual query of the GPT models and the challenges of the additional RL-related operations.  I agree with the overall attitude of the authors that this particular concern is not very important.  The performance of the graphGA algorithm is nearly identical to that of the new method, despite its extreme simplicity and efficiency.  On the other hand the lack of pre-training is a limitation of graphGA, at least in principle.


**Questions:**

What is the radius and the bitlength in the features in Eq(2).  Is chirality included?

This may be a limitation of the notation used in the paper and perhaps the authors already tried to do the correct thing in the code (I didn't check), but it appears that equation (4) does not use a standardized SMILES representation of the molecule, so it is possible that the agents will try to learn to produce different textual representations of the same molecule (i.e. each agent could canonicalize in a different way).  Of course the magnitude of such an effect probably depends on the details of the training protocol and on the amount of augmentation with randomized SMILES in the pre-training protocol.  Would it be possible to instead use the canonical and certain randomly sampled representations of the same molecule as the inputs x to equation (4)?  Could the authors test their final agents for their preference towards different textual representations of the same molecules?

The statement at the end of page 7 is very unclear: did the authors test the listed small molecules in tables 2 and 3 against the SARS CoV-2 protease and polymerase in a laboratory setting and show that they don't work?  If so, it would be very useful to provide a brief description of these experiments in a supplemental material or cite a separate publication after anonymization is no longer a concern.  At a minimum, they should clarify the language at the end of that section.

Regarding performance, did the authors monitor the GPU usage during training and do they achieve the maximal power draw on the GPU during the optimization experiments?


**Limitations:**

This work has no negative societal impacts.

---

> ### Author Rebuttal · Authors · 2023-08-07
>
> Thanks for your approval and valuable review comments! We address your main concerns below:
>
> **Q1 (in *weaknesses*)**: A minor weakness of this method is that the performance of the code is somewhat slow, which is somewhat understandable given the slow individual query of the GPT models and the challenges of the additional RL-related operations.
>
> **A1**: While our algorithm does require a considerable amount of time to execute, the primary time consumption does not stem from the inference and updating of GPT agents. Instead, the bulk of the time is dedicated to the computation of oracles on CPUs. For instance, in a single docking task for a specific target, the entire reinforcement learning process demands around 100 hours when executed on a single NVIDIA A100 GPU alongside 64 CPU cores, during which more than 95% of the time is cost by the docking process itself, even though parallel computing of Quick Vina software is implemented. Therefore, the issue of slow drug design isn't unique to our algorithm; other methods also struggle with low efficiency when dealing with time-intensive oracles like docking. Furthermore, as you rightly mentioned, the runtime of our algorithm remains significantly shorter when compared to the years that conventional drug discovery processes typically span.
>
> **Q2 (in *weaknesses*)**: The performance of the GraphGA algorithm is nearly identical to that of the new method, despite its extreme simplicity and efficiency. On the other hand the lack of pre-training is a limitation of graphGA, at least in principle.
>
> **A2**: Initially, it's important to acknowledge that although GraphGA does not leverage the deep learning techniques that have gained prominence in recent times, it has been proven to be a competitive approach for molecular design. It ranks second in a comprehensive benchmark test, trailing only behind another SMILES-based reinforcement learning method  (https://openreview.net/forum?id=yCZRdI0Y7G). Nevertheless, as you mentioned, this approach may have approached the zenith of performance attainable through non-deep methods. This implies that for intricate molecular generation tasks, such as docking, GraphGA's performance is likely to be suboptimal, because it cannot effectively utilize the chemical information embedded within molecular datasets through pre-training paradigm.
>
> **Q3 (in *questions*)**: What is the radius and the bitlength in the features in Eq(2). Is chirality included?
>
> **A3**: For the implementation of internal diversity, we adopt the function in the widely-used Therapeutics Data Commons (TDC) package (https://github.com/mims-harvard/TDC/blob/6af2a41679a0699446ad627be8051504548e86fa/tdc/chem_utils/evaluator.py#L99C31-L99C31). Specifically, in the Morgan fingerprints (ECFPs) the radius is 2, the bitlength is 2048, and chirality is not included.
>
> **Q4 (in *questions*)**: It appears that equation (4) does not use a standardized SMILES representation of the molecule, so it is possible that the agents will try to learn to produce different textual representations of the same molecule. Would it be possible to instead use the canonical and certain randomly sampled representations of the same molecule as the inputs x to equation (4)? Could the authors test their final agents for their preference towards different textual representations of the same molecules?
>
> **A4**: We refrain from employing the canonicalized SMILES representation while computing the generation probability in equations (3) and (4). This choice arises from the fact that GPT agents inherently grasp the grammar of SMILES strings rather than focusing on molecular structures. The probability using an equivalent yet non-originally generated SMILES string lacks a direct meaning in reinforcement learning optimization. As a result, it is indeed possible for the agents to produce different SMILES of the same molecule.
>
> Nevertheless, when computing property scores, the variability in SMILES representation does not influence the outcome, since oracles take as input either the molecule items in RDKit or molecular fingerprints, both of which uniquely correspond to the molecular structure. Furthermore, in the process of updating the molecular memory, canonicalized SMILES are employed. This ensures that a given molecule does not exist in the memory in multiple forms, preventing duplicate entries.
>
> **Q5 (in *questions*)**: The statement at the end of page 7 is very unclear: did the authors test the listed small molecules in tables 2 and 3 against the SARS CoV-2 protease and polymerase in a laboratory setting and show that they don't work?
>
> **A5**: We sincerely regret any confusion caused. Our approach lacked wet experiments to validate the molecular properties outlined in tables 2 and 3 due to the absence of experimental conditions. The values presented in the tables are the result of in silico oracles. The intention of our statement at the end of page 7 is to emphasize that:
>
> As the Quantitative Estimate of Drug-Likeness (QED) is calculated through a comparison with the distribution of compounds in the existing drug database – one that lacks molecules designed for SAR-COV-2 – it's important to note that a low QED score doesn't necessarily imply inefficacy against the SAR-COV-2 targets. On the contrary, molecules exhibiting effectiveness might well lie beyond the boundaries of the existing drug compound distribution.
>
> **Q6 (in *questions*)**: Regarding performance, did the authors monitor the GPU usage during training and do they achieve the maximal power draw on the GPU during the optimization experiments?
>
> **A6**: As we explained in **A1**, the predominant portion of our algorithm's execution time is cost by the oracles running on the CPU, despite utilizing 64 CPU cores. Consequently, the GPU operated below its maximal capacity throughout the reinforcement learning process and is barely operating most of the time in the docking tasks.

---

> > ### Comment · Reviewer_4Wce · 2023-08-18
> > **Thanks for the reply**
> >
> > I acknowledge having read the response by the authors.  I am a little skeptical that the answer A4 is sufficient to introduce substantial diversity given the easy choice of randomizing the string representation of the best molecule.  I wonder if the diversity result is severely suboptimal compared to what such a method could achieve.

---

> > > ### Author Response · Authors · 2023-08-19
> > >
> > > Thanks for your reply! In response to your question, the randomness of SMILES itself does not affect the molecular diversity measurements, because internal diversity (equation (2)) is calculated for molecules, not SMILES strings. Even if multiple different SMILES strings of the same molecule are generated, it will only be counted once in internal diversity.

---

### Official Review · Reviewer_xd9X · 2023-07-25

**Soundness:** 2 fair
**Presentation:** 2 fair
**Contribution:** 3 good
**Rating:** 4
**Confidence:** 3

**Summary:**

This paper proposes a novel multi-agent reinforcement learning algorithm with agents parameterized with a pre-trained GPT architecture for de novo drug design.  The authors propose a modified objective function with an intrinsic reward inspired bonus to encourage diversity between agents and also propose to use a constraint to keep the fine-tuned agents close to the pre-trained agents.  The authors evaluate their algorithm on Guacamol, by generating a number of inhibitors for SARS-CoV2 targets.  They also perform ablations on their method with the GSK3$\beta$ and JNK3 maximization tasks.

**Strengths:**

Some strengths of this paper are:

- An intrinsic reward like term added to the agents' loss which encourages diversity.
- The method performs favorably to the other methods compared against on Guacamol benchmark.
- It performs comprehensive ablations on the GSK3$\beta$ and JNK3 maximization tasks

**Weaknesses:**

My main complaint about this paper is that I am not convinced of the algorithm's superiority over rivaling methods based upon the experiments section.  I feel that the paper has both missed some necessary baseline methods and that the experiments as they stand are insufficient in demonstrating the paper's main claim that its method leads to improved diversity over other methods.  I also have concerns about missing related work in multi-agent RL in which there is already a body of literature on encouraging diversity among agents, as well as other competing methods which have been applied to molecular drug generation such as diffusion models and GFlowNets.  I also have concerns regarding this paper's reproducibility.  I will go over these concerns one by one.

## Experiments section
1. The main claim of this paper is that its approach leads to superior diversity, but I did not see any experiments _comparing_ the diversity of molecules it generates to molecules generated by other methods.  Indeed, there were experiments looking at the diversity of molecules generated by their method, but it was only their method.  There isn't an indication whether their main claim of improved diversity is true if there is no comparison to other methods.
2. While the generated molecules do look diverse to my non-chemist eye, I would like to see more generated molecules and critically some measurement of diversity of the generated molecules to be convinced.  Also, these results were on only one seed which is insufficient.  At minimum there should be three seeds, ideally quite a bit more (see [here](https://ai.googleblog.com/2021/11/rliable-towards-reliable-evaluation.html)).
3. There are missing baselines in the experiments.  The authors should have compared to an existing LLM molecule generation method such as MolGPT, but this was missing from their experiments.  There are other methods for encouraging diversity for molecule design with RL-inspired machinery such as GFlowNets (https://arxiv.org/abs/2106.04399) that should be compared against.  Also, it would have been nice to see a comparison with one of the recent works on diffusion for molecular design (e.g., https://arxiv.org/pdf/2203.17003.pdf, https://arxiv.org/pdf/2305.01140.pdf), or at minimum a compelling reason for why not to compare to these methods.
4. In the Guacamol experiments it seems experiments were run over one seed.  This is insufficient.
5. A more minor comment: it's hard to understand the significance of the Guacamol task when the tasks are ordered 1-20 without context of what the tasks actually are.

## Missing references
1. I mentioned in the last point, but it would have been nice to see some discussion of other methods such as GFlowNets or diffusion models which also try to encourage diversity in molecular design.
2. There is already a body of literature on encouraging diversity in multi-agent RL, but I did not see references to this literature.  Some representative papers may be https://arxiv.org/abs/2106.02195 and https://openreview.net/forum?id=H-6iczs__Ro.

## Reproducibility
1. All experiments seem to have been run with one seed.  There is no way to know if the results would hold with more seeds.
2. There is no listing of the hyperparameters used or the hyperparameter tuning methods used (or values tried if a grid search).
3. There is no (anonymized) submitted code available to verify or reproduce the authors' claims.

## Clarity issues
1. In the section explaining the loss function, the indexing used is rather confusing.  E.g., the authors use a loss $L_1$ which seems fixed, then also a loss $L_k$ which seems to index the different agents (so what about when $k=1$?).

**Questions:**

1. In the loss function, why does the sum over agents only go up to $k-1$?  Shouldn't the second loss term be something like $\sum_{j=1}^n s(x) \left|P_k(x) - P_j(x)\right|$?  Why compare to only the agents before this index as the ordering seems arbitrary.
2. Did the authors consider using a method for encouraging diversity which already exists in the multi-agent RL literature?  If so, why did they not use it and did they run any comparisons?

**Limitations:**

The main and most important limitation of this work is that I do not know whether the proposed method actually is competitive with rival methods due to some missing experiments, baselines, and insufficient reproducibility.

---

> ### Author Rebuttal · Authors · 2023-08-08
>
> Thanks for your valuable review comments! We address your main concerns below, and we sincerely hope that you will reconsider and upgrade your rating:
>
> **Q1 (diversity comparison, in *experiments 1, 2*)**
>
> **A1**: To further demonstrate the advantages in diversity of our approach, we have added several baselines to the experiments on GSK3$\beta$ and JNK3 maximization (**Table 2 in pdf**)  for comparison. Additionally, we supplement a experiment on QED maximization (**Table 3 in pdf**) with diversity measurements. The results of these experiments indicate that our approach is indeed effective in promoting the molecular diversity in drug design.
>
> We did not provide diversity measurements in experiments on the GuacaMol benchmark, where each task produces a single score that comprehensively represents the performance of a given method (following the official guidelines). Therefore, the advantages of our approach in terms of diversity have been inherently contained in our higher scores, while reporting molecular diversity in GuacaMol is unreasonable.
>
> **Q2 (missing baselines, in *experiments 3 / missing references 1*)**
>
> **A2**:
>
> 1. We have listed the related LLM-based methods of molecular generation in our paper (line 92), but notably none of them are designed for *de novo* drug design tasks. It should be especially discriminated that *de novo* drug design aims to generate molecules with properties beyond the property distribution of the training set, while previous LLM-based methods cannot achieve this  (https://arxiv.org/abs/2203.14500). Specifically, MolGPT, mentioned by the reviewer, is a pre-trained model aiming to learn the existing datasets and generate molecules within the property distribution. Other LLM-based approaches, including ChemFormer, also do not target *de novo* drug design, so we did not include them in our experiments.
> 2. You also mention some diffusion-based methods for molecular generation, but at the moment diffusion model works only in 3D molecular generation, which mainly target the structure and quantum properties. In contrast, 1D/2D molecular generation (our aim) focuses on the biochemical properties, which is a different direction of research than 3D molecular generation (https://arxiv.org/abs/2203.14500). Hence, we did not include diffusion models in our experiments.
> 3. GFlowNet is a baseline of *de novo* drug design, and we didn't include it in the original paper mainly because of its relatively poor performance. As supplementary, we have included its results in **Table 1, 2, 3 (in pdf)**.
>
> **Q3 (multiple seeds & Guacamol tasks, in *experiments 2, 4, 5 / reproducibility 1*)**:
>
> **A3**: First, we would like to emphasize that in section 4.3, we report the standard deviations across different seeds. In the GuacaMol benchmark experiment, we have also launched multiple runs to verify the robustness of the algorithm. However, due to the high stability of our algorithm, and most of the previous works on GuacaMol did not report standard deviations, so we did not include them in the original paper.
>
> However, to address any reservations, we have included the standard deviation values for our algorithm's performance on the GuacaMol task across 5 different seeds in **Table 1 (in pdf)**. Additionally, we have listed the names of the 20 GuacaMol tasks, the specifics of which can be found in the GuacaMol paper.
>
> **Q4 (MARL references, in *missing references 2/questions 2*)**
>
> **A4**: Although, as you've mentioned, previous literature in cooperative multi-agent reinforcement learning has presented some approaches to encourage diversity. However, they can hardly be directly applied to *de novo* drug design because:
>
> 1. Typically, the objectives of these techniques are enhancing the behavioral diversity of agents, leading to a collection of diverse agents at the end of the RL process. In contrast, our aim centers on enhancing the diversity of the objects (molecules in our context) that the agents are tasked with searching.
> 2. These methods are typically designed for well-defined virtual spaces (like games) or real-world 3D spaces, leveraging specific space characteristics such as trajectories, which are difficult to define and utilize in the chemical space.
>
> **Q5 (in *reproducibility 2, 3*)**
>
> **A5**: Our codes are available at https://anonymous.4open.science/r/MolRL-MGPT-835E. We explain your question in detail in the author rebuttal.
>
> **Q6 (in *clarity*)**
>
> **A6**: We have accurately formulated the loss function in section 3.2. Specifically, Eq. (3) defines the loss function for the first agent, which is also the first term of the loss functions for other agents. Eq. (4) defines the loss functions for all agents, and it's noteworthy that when $k=1$, Eq. (4) simplifies to Eq. (3).
>
> **Q7 (*questions 1*)**
>
> **A7**: For the design of loss functions, a seemingly natural approach is to reward the differences between all agents equally. However, this method exhibits relatively low robustness in experiments, that is, agents are likely to greatly lose the validity of generated SMILES due to great mutual interference. In contrast, our approach (Eq. (4)) organizes agents in a sequence, and each agent is only rewarded for the difference between it and the agents preceding it, which enables agents to get promoted more stably.

---

> > ### Comment · Reviewer_xd9X · 2023-08-14
> >
> > Thanks to the authors for their work on improving their paper!  I have a few comments to their response, which I list below.
> >
> > ### On Q1
> > I appreciate your adding diversity metrics to the JNK3 and GSK3 maximization tasks and agree that this betters the case for this paper.  I note that it some measure of diversity on the COVID target synthesis task would've been appreciated as well.
> >
> > ### On Q2
> > > We have listed the related LLM-based methods of molecular generation in our paper (line 92), but notably none of them are designed for de novo drug design tasks. It should be especially discriminated that de novo drug design aims to generate molecules with properties beyond the property distribution of the training set, while previous LLM-based methods cannot achieve this (https://arxiv.org/abs/2203.14500). Specifically, MolGPT, mentioned by the reviewer, is a pre-trained model aiming to learn the existing datasets and generate molecules within the property distribution. Other LLM-based approaches, including ChemFormer, also do not target de novo drug design, so we did not include them in our experiments.
> >
> > The authors' argument here is fair, though I will note that the same argument could be made about JT-VAE which the authors do include as a baseline.  There, the VAE is used with a Bayesian optimization procedure, and a similar setup could be applied to the other LLM-based approaches, though I agree that this may be outside the scope of this paper (but if the authors did this and showed their method outperforms it would certainly improve the case for their method!).
> >
> > > You also mention some diffusion-based methods for molecular generation, but at the moment diffusion model works only in 3D molecular generation, which mainly target the structure and quantum properties. In contrast, 1D/2D molecular generation (our aim) focuses on the biochemical properties, which is a different direction of research than 3D molecular generation (https://arxiv.org/abs/2203.14500). Hence, we did not include diffusion models in our experiments.
> >
> > I'm not sure what is included in 1D/2D molecular generation, but there are graph based diffusion models for molecule generation, e.g., https://arxiv.org/abs/2209.14734.  You can see https://arxiv.org/pdf/2304.01565.pdf for a survey on more of these sorts of methods.  It's fair that the main focus of this paper is on LLM based approaches to de novo drug design, but since these 3D based methods seem to work well I think including one as a baseline would be helpful to illustrate the benefit of the authors' approach.
> >
> > > GFlowNet is a baseline of de novo drug design, and we didn't include it in the original paper mainly because of its relatively poor performance. As supplementary, we have included its results in Table 1, 2, 3 (in pdf).
> >
> > Great! :)
> >
> > ### On Q3
> > > First, we would like to emphasize that in section 4.3, we report the standard deviations across different seeds. In the GuacaMol benchmark experiment, we have also launched multiple runs to verify the robustness of the algorithm. However, due to the high stability of our algorithm, and most of the previous works on GuacaMol did not report standard deviations, so we did not include them in the original paper.
> >
> > It's indeed unfortunate that prior works did not run experiments over multiple seeds.  It's good to see the authors running their experiments with more seeds on the GuacaMol experiment.  I'm curious what the results across multiple seeds are for the baselines for GuacaMol as I only see multiple seeds for the MARL experiments.
> >
> > I would also like to see performance vs other baselines on the COVID task, again across other seeds to consider raising my score (e.g., report values of top-k across 1k or some similar number of molecules generated per seed, or mean of the reported metrics across the 1k generated molecules per seed).
> >
> > > Additionally, we have listed the names of the 20 GuacaMol tasks, the specifics of which can be found in the GuacaMol paper.
> >
> > Great! :)
> >
> > ### On Q4
> > > Typically, the objectives of these techniques are enhancing the behavioral diversity of agents, leading to a collection of diverse agents at the end of the RL process. In contrast, our aim centers on enhancing the diversity of the objects (molecules in our context) that the agents are tasked with searching.
> >
> > I don't see why behavioral diversity of agents wouldn't also lead to improved diversity of the objects generated.  Shouldn't more diverse behavior policies generate more diverse objects?  If not, why not?
> >
> > > These methods are typically designed for well-defined virtual spaces (like games) or real-world 3D spaces, leveraging specific space characteristics such as trajectories, which are difficult to define and utilize in the chemical space.
> >
> > Why are trajectories difficult to use in chemical space?  In the case of this paper they should just be the generated SMILES string, no?

---

> > > ### Comment · Reviewer_xd9X · 2023-08-14
> > >
> > > ### On Q5
> > > Thanks for the code, and that's really unfortunate that the system was down, I see that and adjust my impression to account for it.  However, to consider adjusting my score the paper still needs to improve its reproducibility.  E.g., the hyperparameter tuning method used for tuning baselines and the authors' method (and parameter values tried if a grid search) are not provided, nor are the actual hyperparameter values selected for each method.  The number of seeds for each experiment should be consistent and listed in one place in the paper.  It would be nice (but not necessary) to have the hardware and time used to train for each experimental setting (GuacaMol, COVID targets, and maximization tasks) and algorithm tried (this paper's and that used when running the baselines).
> > >
> > > ### On Q6
> > > Got it!
> > >
> > > ### On Q7
> > > That's an interesting observation and I'd encourage the authors to discuss it more in the paper (or in Appendix).  I'm curious why this would take place as the intuition is not clear to me.
> > >
> > > Overall I very much appreciate the authors' response.  My main issue which I feel is still unresolved is the reproducibility of this study, as the hyperparameters tuned over are not listed, different seeds are not existing yet for the baselines in the GuacaMol experiments, and no other baselines were tried on the Covid task.  I'm also still a bit confused about the authors' explanations about why their method is superior as opposed to other methods focusing on MARL agent diversity.  Without multiple seeds for all experiments and baselines and baselines being run for the COVID experiments, I cannot consider improving my score.

---

> > > > ### Author Response · Authors · 2023-08-19
> > > >
> > > > Thanks for your valuable review comments! To address your concerns, we have added more experimental results and details below, as well as further explanation of your questions. Due to the length limit of comments, our responses consist of multiple parts.
> > > >
> > > > We sincerely hope that you will reconsider and upgrade your rating of our paper.
> > > >
> > > > **Part 1: Adding baselines to the docking tasks against SAR-COV2 targets (On Q1, Q3)**
> > > >
> > > > In experiments of designing inhibitors against two protein targets for SARS-CoV2, we add baselines (with 5 different random seeds) for comparison, and report the mean scores and diversity of the top-100 molecules. The results indicate that:
> > > >
> > > > 1. JT-VAE and GFlowNet cannot work on docking tasks, as their generated molecules are of bad docking scores (none of molecules have docking scores less than -10).
> > > > 2. Compared with all other baselines, our **MolRL-MGPT** achieves better mean docking scores and internal diversity, as well as similar QED and SA scores.
> > > >
> > > > So this experiment shows the advantages of our approach again.
> > > >
> > > > **PLPro_7JIR:** (top-100 molecules, 5 different random seeds)
> > > >
> > > > |    Methods     | Mean docking score (*↓*) | Mean QED score (*↑*) | Mean SA score (*↓*) |      IntDiv       |
> > > > | :------------: | :----------------------: | :------------------: | :-----------------: | :---------------: |
> > > > |     JT-VAE     |       -8.76 ± 0.35       |    0.795 ± 0.038     |    2.994 ± 0.140    |   0.836 ± 0.032   |
> > > > |    GFlowNet    |       -9.11 ± 0.21       |    0.726 ± 0.015     |    2.823 ± 0.076    |   0.825 ± 0.010   |
> > > > |    GraphGA     |      -10.83 ± 0.08       |    0.380 ± 0.013     |    3.638 ± 0.162    |   0.740 ± 0.017   |
> > > > |    Reinvent    |      -10.75 ± 0.05       |    0.392 ± 0.008     |    2.649 ± 0.035    |   0.619 ± 0.023   |
> > > > | **MolRL-MGPT** |    **-11.02 ± 0.06**     |  **0.386 ± 0.006**   |  **2.550 ± 0.047**  | **0.745 ± 0.008** |
> > > >
> > > > **RdRp_6YYT:** (top-100 molecules, 5 different random seeds)
> > > >
> > > > |    Methods     | Mean docking score (*↓*) | Mean QED score (*↑*) | Mean SA score (*↓*) |      IntDiv       |
> > > > | :------------: | :----------------------: | :------------------: | :-----------------: | :---------------: |
> > > > |     JT-VAE     |       -8.33 ± 0.25       |    0.719 ± 0.019     |    2.959 ± 0.094    |   0.828 ± 0.018   |
> > > > |    GFlowNet    |       -8.89 ± 0.16       |    0.656 ± 0.033     |    2.854 ± 0.061    |   0.770 ± 0.015   |
> > > > |    GraphGA     |      -11.26 ± 0.12       |    0.262 ± 0.010     |    3.520 ± 0.049    |   0.658 ± 0.009   |
> > > > |    Reinvent    |      -11.30 ± 0.04       |    0.275 ± 0.006     |    2.917 ± 0.035    |   0.616 ± 0.021   |
> > > > | **MolRL-MGPT** |    **-11.84 ± 0.07**     |  **0.278 ± 0.005**   |  **2.894 ± 0.072**  | **0.670 ± 0.013** |

---

> > > > > ### Author Response · Authors · 2023-08-19
> > > > >
> > > > > **Part 2: Multiple seeds for all baselines in the GuacaMol benchmark (On Q3)**
> > > > >
> > > > > In the GuacaMol benchmark, we run each baseline over 5 different seeds to demonstrate the steady advantages of our approach. Note that there is no randomness in the results of the ChEMBL dataset.
> > > > >
> > > > > |            Tasks            | ChEMBL dataset |  Graph MCTS   |   SMILES GA    |  SMILES LSTM   |    Graph GA    |    Reinvent    |      GEGL      |   GFlowNet    |     MolRL-MGPT     |
> > > > > | :-------------------------: | :------------: | :-----------: | :------------: | :------------: | :------------: | :------------: | :------------: | :-----------: | :----------------: |
> > > > > |  1. Celecoxib rediscovery   |     0.505      | 0.357 ± 0.004 | 0.730 ± 0.005  | 1.000 ± 0.000  | 1.000 ± 0.000  | 1.000 ± 0.000  | 1.000 ± 0.000  | 0.412 ± 0.012 | **1.000 ± 0.000**  |
> > > > > | 2. Troglitazone rediscovery |     0.419      | 0.310 ± 0.015 | 0.519 ± 0.010  | 1.000 ± 0.000  | 1.000 ± 0.000  | 1.000 ± 0.000  | 0.557 ± 0.004  | 0.230 ± 0.028 | **1.000 ± 0.000**  |
> > > > > | 3. Thiothixene rediscovery  |     0.456      | 0.315 ± 0.006 | 0.590 ± 0.016  | 1.000 ± 0.000  | 1.000 ± 0.000  | 1.000 ± 0.000  | 1.000 ± 0.000  | 0.335 ± 0.009 | **1.000 ± 0.000**  |
> > > > > | 4. Aripiprazole similarity  |     0.595      | 0.392 ± 0.018 | 0.834 ± 0.003  | 1.000 ± 0.000  | 1.000 ± 0.000  | 1.000 ± 0.000  | 1.000 ± 0.000  | 0.593 ± 0.015 | **1.000 ± 0.000**  |
> > > > > |   5. Albuterol similarity   |     0.719      | 0.742 ± 0.010 | 0.908 ± 0.002  | 1.000 ± 0.000  | 1.000 ± 0.000  | 1.000 ± 0.000  | 1.000 ± 0.000  | 0.467 ± 0.023 | **1.000 ± 0.000**  |
> > > > > |   6. Mestranol similarity   |     0.629      | 0.402 ± 0.002 | 0.795 ± 0.009  | 1.000 ± 0.000  | 1.000 ± 0.000  | 1.000 ± 0.000  | 1.000 ± 0.000  | 0.401 ± 0.007 | **1.000 ± 0.000**  |
> > > > > |          7. C11H24          |     0.684      | 0.421 ± 0.033 | 0.826 ± 0.004  | 0.993 ± 0.001  | 0.970 ± 0.001  | 0.999 ± 0.000  | 1.000 ± 0.000  | 0.532 ± 0.010 | **1.000 ± 0.000**  |
> > > > > |      8. C9H10N2O2PF2Cl      |     0.747      | 0.652 ± 0.025 | 0.890 ± 0.007  | 0.882 ± 0.005  | 0.981 ± 0.003  | 0.878 ± 0.005  | 1.000 ± 0.000  | 0.228 ± 0.012 | **0.939 ± 0.003**  |
> > > > > |    9. Median molecules 1    |     0.334      | 0.217 ± 0.016 | 0.340 ± 0.012  | 0.432 ± 0.005  | 0.407 ± 0.001  | 0.432 ± 0.003  | 0.454 ± 0.002  | 0.210 ± 0.020 | **0.447 ± 0.006**  |
> > > > > |   10. Median molecules 2    |     0.351      | 0.172 ± 0.003 | 0.378 ± 0.002  | 0.424 ± 0.003  | 0.432 ± 0.002  | 0.395 ± 0.001  | 0.435 ± 0.004  | 0.196 ± 0.006 | **0.423 ± 0.004**  |
> > > > > |     11. Osimertinib MPO     |     0.839      | 0.888 ± 0.009 | 0.887 ± 0.005  | 0.915 ± 0.011  | 0.956 ± 0.006  | 0.887 ± 0.008  | 1.000 ± 0.000  | 0.791 ± 0.005 | **0.977 ± 0.001**  |
> > > > > |    12. Fexofenadine MPO     |     0.817      | 0.703 ± 0.012 | 0.942 ± 0.018  | 0.960 ± 0.002  | 0.997 ± 0.000  | 1.000 ± 0.000  | 1.000 ± 0.000  | 0.723 ± 0.014 | **1.000 ± 0.000**  |
> > > > > |     13. Ranolazine MPO      |     0.792      | 0.602 ± 0.014 | 0.884 ± 0.004  | 0.852 ± 0.005  | 0.915 ± 0.008  | 0.895 ± 0.004  | 0.930 ± 0.005  | 0.675 ± 0.017 | **0.939 ± 0.000**  |
> > > > > |     14. Perindopril MPO     |     0.575      | 0.375 ± 0.020 | 0.650 ± 0.013  | 0.806 ± 0.003  | 0.794 ± 0.005  | 0.766 ± 0.004  | 0.833 ± 0.001  | 0.459 ± 0.011 | **0.809 ± 0.005**  |
> > > > > |     15. Amlodipine MPO      |     0.696      | 0.537 ± 0.005 | 0.731 ± 0.018  | 0.891 ± 0.006  | 0.891 ± 0.004  | 0.884 ± 0.009  | 0.904 ± 0.001  | 0.427 ± 0.006 | **0.906 ± 0.001**  |
> > > > > |     16. Sitagliptin MPO     |     0.509      | 0.458 ± 0.009 | 0.692 ± 0.007  | 0.557 ± 0.014  | 0.890 ± 0.001  | 0.538 ± 0.025  | 0.752 ± 0.008  | 0.048 ± 0.007 | **0.822 ± 0.003**  |
> > > > > |      17. Zaleplon MPO       |     0.547      | 0.484 ± 0.006 | 0.419 ± 0.010  | 0.663 ± 0.008  | 0.755 ± 0.003  | 0.591 ± 0.001  | 0.764 ± 0.003  | 0.069 ± 0.005 | **0.790 ± 0.008**  |
> > > > > |    18. Valsartan SMARTS     |     0.259      | 0.050 ± 0.013 | 0.555 ± 0.007  | 0.978 ± 0.001  | 0.989 ± 0.002  | 0.102 ± 0.016  | 1.000 ± 0.000  | 0.000 ± 0.000 | **0.997 ± 0.000**  |
> > > > > |        19. deco hop         |     0.933      | 0.592 ± 0.002 | 0.969 ± 0.002  | 0.996 ± 0.000  | 1.000 ± 0.000  | 0.992 ± 0.002  | 1.000 ± 0.000  | 0.605 ± 0.029 | **1.000 ± 0.000**  |
> > > > > |      20. scaffold hop       |     0.738      | 0.465 ± 0.019 | 0.887 ± 0.003  | 0.998 ± 0.000  | 1.000 ± 0.000  | 0.990 ± 0.001  | 1.000 ± 0.000  | 0.484 ± 0.012 | **1.000 ± 0.000**  |
> > > > > |          **Total**          |     12.144     | 9.134 ± 0.046 | 14.426 ± 0.033 | 17.347 ± 0.008 | 17.977 ± 0.005 | 16.349 ± 0.022 | 17.627 ± 0.002 | 7.885 ± 0.067 | **18.049 ± 0.003** |

---

> > > > > > ### Author Response · Authors · 2023-08-19
> > > > > >
> > > > > > **Part 3: Reproducibility: hyperparameters and resource consumption (On Q5)**
> > > > > >
> > > > > > We have provided some experimental details in the experimental section of the original paper, including hyper-parameter settings, and hardware and time consumptions. Here we give further details.
> > > > > >
> > > > > > 1. **On GuacaMol benchmark:**
> > > > > >
> > > > > >    Related details in **line 205** in our paper:
> > > > > >
> > > > > >    > The prior model for MolRL-MGPT was pretrained on the official GuacaMol dataset, which is a subset of ChEMBL. The hyper-parameters are set such that we run each tasks for 5000 iterations (break if the score has achieved 1.000) with 4 GPT agents, and the batch size (number of sampled SMILES strings) of each agent is 256. The values of coefficients are set to $\sigma_1=1000$ with a linear decreasing schedule, and $\sigma_2=0.1$. The entire set of 20 tasks tasks less than 400 hours to complete when run on a single NVIDIA A100 GPU.
> > > > > >
> > > > > >    This includes all the pivotal hyper-parameter settings in this experiment.
> > > > > >
> > > > > > 2. **On SARS-COV2 targets:**
> > > > > >
> > > > > >    Related details in **line 241** in our paper:
> > > > > >
> > > > > >    > We run the RL process for 1000 iterations on each target with 4 GPT agents, and the batch size of each agent is 128. The whole process for one target takes approximately 100 hours on a single NVIDIA A100 GPU and 64 CPU cores. The details of generated candidates are as follows.
> > > > > >
> > > > > >    The coefficients remain to be $\sigma_1=1000$ with a linear decreasing schedule, and $\sigma_2=0.1$.
> > > > > >
> > > > > > 3. **On GSK3$\beta$, JNK3 and QED maximization:**
> > > > > >
> > > > > >    Related details in **line 276** in our paper:
> > > > > >
> > > > > >    > We run the RL process for 1000 iterations in every experiment, and the total batch size of all agents is 256. Each process takes less than 2 hours on a single NVIDIA A100 GPU.
> > > > > >
> > > > > >    The coefficients remain to be $\sigma_1=1000$ with a linear decreasing schedule, and $\sigma_2=0.1$. The numbers of agents are listed in the table of experimental results.
> > > > > >
> > > > > > 4. **For other baselines:** We all adopt the implementations and settings by https://openreview.net/forum?id=yCZRdI0Y7G, which is a comprehensive benchmark of methods for *de novo* 1D/2D molecular generation. We promise to contain these details in the final version of our paper.
> > > > > >
> > > > > > 5. **For hyper-parameter tuning:** Our algorithm is not very sensitive to hyper-parameter settings, so we did not conduct a grid search. Some experiments for selecting parameters are conducted in GSK3$\beta$ and JNK3 maximization. For further details, please refer to Q4/A4 in my rebuttal to Reviewer AzpA.
> > > > > >
> > > > > > 6. **For multiple seeds:** The standard deviations in all experiments are calculated by running across 5 different seeds.

---

> > > > > > > ### Author Response · Authors · 2023-08-19
> > > > > > >
> > > > > > > **Part 4: Further clarification of related works (On Q2/Q4)**
> > > > > > >
> > > > > > > 1. About **previous LLM-based methods**: There may be ways to transfer previous pre-trained LLMs to *de novo* drug design, but this is a completely new research work that requires careful design and a lot of experiments. As you said, this is outside the scope of our paper.
> > > > > > >
> > > > > > > 2. About **diffusion-based methods**: We reiterate that our method is designed for ***de novo* 1D/2D molecular generation**. The two diffusion-based works you mentioned in the original review (https://arxiv.org/pdf/2203.17003.pdf, https://arxiv.org/pdf/2305.01140.pdf) are both for **3D molecular generation**, and DiGress (https://arxiv.org/abs/2209.14734), which you mentioned in the new comments is for **non-*de novo* 1D/2D molecular generation**.
> > > > > > >
> > > > > > >    The categories of the methods can be directly recognized from the experiment tasks, and because of their different application scenarios, we cannot use these diffusion-based methods as baselines in our paper. By the way, for specific classification of molecular generation tasks, including "1D/2D vs. 3D", and "*de novo* vs. non-*de novo*", please refer to the survey https://arxiv.org/abs/2203.14500.
> > > > > > >
> > > > > > > 3. About **MARL related works**: Previous literature in cooperative multi-agent reinforcement learning has presented some approaches to encourage diversity. However, they can hardly be directly applied to *de novo* drug design because:
> > > > > > >
> > > > > > >    - Typically, the objectives of these techniques are enhancing the behavioral diversity of agents, leading to a collection of **diverse agents at the end of the RL process**. In contrast, our aim centers on enhancing the **diversity of the molecules generated by agents during the RL process**. These are two completely distinct objectives, as finally diverse agents do not necessarily generate diverse molecules in the previous process. (https://openreview.net/forum?id=H-6iczs__Ro, https://openreview.net/forum?id=lvRTC669EY_)
> > > > > > >    - These methods are typically designed for well-defined virtual spaces (like games) or real-world 3D spaces, leveraging specific space characteristics such as trajectories, which are difficult to define and utilize in the chemical space, since **there is no coordinates or grids defined in the chemical space. SMILES strings cannot be directly converted to computable spatial positions.** (https://arxiv.org/abs/2106.02195)
> > > > > > >
> > > > > > > However, since our approach seems likely to be relevant to the diffusion-based methods and MARL literature, we promise to discuss these in the final version of our paper.

---

### Author Rebuttal · Authors · 2023-08-08

We would like to express our sincere appreciation to all the reviewers for your valuable feedback on our paper, and we have responded to all your questions (in the corresponding rebuttal sections). We also add some supplementary experimental results in the **pdf**, including more baselines on the GuacaMol benchmark, more baselines on GSK3$\beta$ and JNK3 maximization, new experiments on QED maximization and other details. We promise to present these contents in the final version of the paper.

In addition, we would like to explain the issue raised by reviewer xd9X that **our codes are not available**:

Indeed, in the abstract of my paper, I have included an Anonymous GitHub link to the MolRL-MGPT codebase (https://anonymous.4open.science/r/MolRL-MGPT-835E).   Regrettably, it appears that during your review, the Anonymous GitHub service encountered an interruption, obstructing your reproducing our experiments and potentially tarnishing your impression.   We sincerely apologize for this inconvenience and want to clarify that we do not consider it to be attributable to our actions.   Now the Anonymous Github has been restored, and the timestamp verifies that we initially committed the code in May.  We earnestly beseech you to reevaluate the code, as it contains many details of our experiments, including hyper-parameter settings, etc.

---

### Decision · Program_Chairs · 2023-09-21

**Decision:**

Accept (poster)

**Comment:**

This paper proposes MolRL-MGPT, a new RL algorithm for de-novo drug design.

The authors had a very productive discussion with the reviewers during the rebuttal period and all the reviewers raised their scores. Specifically, reviewer xd9X changed his mind about the paper after the discussion and new experimental results.

I recommend an accept and encourage the authors to incorporate all the clarifications and additional results to the camera-ready version of the paper.